# Identification and Functional Characterisation of Two Oat UDP-Glucosyltransferases Involved in Deoxynivalenol Detoxification

**DOI:** 10.3390/toxins14070446

**Published:** 2022-06-30

**Authors:** Alfia Khairullina, Nikos Tsardakas Renhuldt, Gerlinde Wiesenberger, Johan Bentzer, David B. Collinge, Gerhard Adam, Leif Bülow

**Affiliations:** 1Division of Pure and Applied Biochemistry, Lund University, 221 00 Lund, Sweden; nikos.tsardakas_renhuldt@tbiokem.lth.se (N.T.R.); johan.bentzer@tbiokem.lth.se (J.B.); leif.bulow@kilu.lu.se (L.B.); 2Department of Plant and Environmental Sciences, University of Copenhagen, 1871 Frederiksberg, Denmark; dbc@plen.ku.dk; 3Institute of Microbial Genetics, Department of Applied Genetics and Cell Biology, University of Natural Resources and Life Sciences, Vienna, Konrad Lorenz Str. 24, 3430 Tulln, Austria; gerlinde.wiesenberger@boku.ac.at (G.W.); gerhard.adam@boku.ac.at (G.A.)

**Keywords:** oats, deoxynivalenol, UDP-glucosyltransferase, glycosylation

## Abstract

Oat is susceptible to several Fusarium species that cause contamination with different trichothecene mycotoxins. The molecular mechanisms behind Fusarium resistance in oat have yet to be elucidated. In the present work, we identified and characterised two oat UDP-glucosyltransferases orthologous to barley HvUGT13248. Overexpression of the latter in wheat had been shown previously to increase resistance to deoxynivalenol (DON) and nivalenol (NIV) and to decrease disease the severity of both Fusarium head blight and Fusarium crown rot. Both oat genes are highly inducible by the application of DON and during infection with *Fusarium graminearum*. Heterologous expression of these genes in a toxin-sensitive strain of *Saccharomyces cerevisiae* conferred high levels of resistance to DON, NIV and HT-2 toxins, but not C4-acetylated trichothecenes (T-2, diacetoxyscirpenol). Recombinant enzymes AsUGT1 and AsUGT2 expressed in *Escherichia coli* rapidly lost activity upon purification, but the treatment of whole cells with the toxin clearly demonstrated the ability to convert DON into DON-3-O-glucoside. The two UGTs could therefore play an important role in counteracting the Fusarium virulence factor DON in oat.

## 1. Introduction

Oat is one of the most important cereals produced worldwide, especially in the temperate regions of Canada, Russia and the Nordic countries [1,2]. The majority of the harvest is used for livestock feed, although its use for human consumption has increased extensively during the last decade due to approved health claims [3,4]. Moreover, the favourable amino acid composition of its proteins, high-lipid, soluble dietary fiber, and bioactive phytonutrients content make oat into an excellent source for innovative plant-based products [5,6]. However, oat is vulnerable to infection by several Fusarium species, which not only damage the kernels and reduce grain weight, and therefore yields, but cause the accumulation of a range of mycotoxins in oat. The most relevant mycotoxins formed by different Fusarium species in oat belong to the large family of trichothecenes [7].

Trichothecenes are classified into groups A and B according to the presence of different substitutions at the C-8 position of the molecule’s backbone [7]. Over the last decade the most prevalent mycotoxins in oat have been deoxynivalenol (DON) (group B trichothecene) and T-2 toxin, together with its hydrolysed form HT-2 toxin (belonging to the highly toxic group A trichothecenes) [8,9,10,11,12,13,14]. DON accumulating in oat is produced mostly by *F. graminearum* [15,16], while the main producers of T-2/HT-2 toxins are *F. langsethiae* and *F. sporotrichoides* [11,17,18,19]. There is a strong tendency for the opposing occurrence of DON and T-2/HT-2 toxins in oat, depending on the weather and agricultural practices [20,21]. Notably, the presence of *F. poae*, which predominantly produces nivalenol (NIV), has been increasing in FHB-infected oat over recent years [19,22,23].

The primary mode of action of trichothecenes is the inhibition of eukaryotic protein synthesis [24,25]. DON, NIV and T-2/HT-2 toxins can cause acute and chronic toxicoses in humans and animals [26,27,28]. Thus, for consumer protection, the European Commission has enacted a maximum tolerated level of DON (1.750 µg/kg for unprocessed oat, and lower levels for food products, for instance, 500 µg/kg in breakfast cereals and 200 µg/kg in processed cereal-based foods and baby foods for infants and young children) [29]. For the sum of T-2/HT-2 toxins (legally nonbinding), indicative levels are 1000 µg/kg for unprocessed oat and 200 µg/kg for oat intended for human consumption, including oat bran and flaked oats [30]. No limits have been set for NIV, but toxicological hazard determination led to the establishment of a tolerable daily intake (TDI) of 1.2 μg/kg body weight per day [31]. Assuming an average oat intake of about 40 g/day in Sweden [32] for a 60 kg person, the TDI would be reached at an NIV contamination level of 1800 µg/kg.

Trichothecenes are also phytotoxic and considered virulence factors of the fungal pathogens. In wheat, DON production by *F. graminearum* is crucial for the efficient spreading of the pathogen from the initially infected spikelet throughout the whole spike [33,34]. Mechanisms counteracting DON (often separated as Type III resistance) lead to increased spreading resistance (type II resistance) [35]. Barley exhibits higher resistance than wheat to pathogen spread, and the loss in trichothecene production consequently has a far less pronounced phenotype. Barley inoculated with a *tri5* mutant of *F. graminearum*, which is unable to produce DON, exhibited lower disease severity and decreased fungal biomass accumulation compared to infection with a wild type [36]. Thus, DON can be inferred to act as a pathogenicity factor even in plants with high type-II resistance.

Plants employ different molecular mechanisms to detoxify mycotoxins. An efficient means to reduce phytotoxicity of trichothecenes is their conjugation with sugars or glutathione (phase II detoxification). While products of both GSH and glucose conjugation are found in wheat grains infected with DON-producing *F. graminearum* [37], DON-3-O-β glucoside (D3G) was found to accumulate to a considerably greater extent [38,39], and glycosylation seems to be the principal detoxification mechanism of DON [37,40]. Whereas the role of DON and its detoxification during the *F. graminearum* infection process had been studied extensively, a contribution of T-2/HT-2 toxins to fungal colonisation, to our knowledge, has not been investigated. Nevertheless, it has been shown in wheat, barley and oat that T-2 toxin can be rapidly metabolised into HT-2, and HT-2-3-O-β-glucoside was found as the main conjugate [41,42,43]. The increased glycosylation of DON with the help of UGTs has been directly linked to the resistance of plants to *F. graminearum* infection. The barley glucosyltransferase *HvUGT13248* is one of the well-studied genes in this respect. The transgenic expression of this barley gene in wheat conferred resistance to both DON and NIV and decreased the disease severity of FHB and Fusarium crown rot (FCR) [44,45]. Moreover, barley lines carrying mutations in the UDP-binding site of HvUGT13248 showed hypersensitivity to DON in a root growth assay and impaired DON to D3G conjugation in spikes [46].

The genetic basis behind the FHB resistance in oat has not been studied in any detail and no trichothecene-detoxifying enzymes are known in oat. Deciphering the molecular mechanisms behind a polygenic trait such as FHB resistance in a plant with a huge hexaploid genome (12.5 Gbp) is a difficult task, particularly in the absence of a genome sequence. Only recently did two newly assembled high resolution whole genome reference sequences of oat *Avena sativa* OT3098 v2 [47] and cv. Sang [48] become available, encouraging our work of finding and characterizing genes potentially important for the breeding of FHB-resistant cultivars. As resistance to trichothecenes is most likely an important component of FHB resistance, the aim of this study was to identify candidate oat UGTs and to test their ability to detoxify trichothecenes produced by Fusarium species on oat.

## 2. Results

### 2.1. Identification of Putative DON-Detoxifying UGTs in Oat

To find oat candidate glucosyltransferases (UGTs) involved in trichothecene detoxification, we searched the proteins present in the orthogroups reported by Kamal et al. [48] using diamond v2.0.0 [49] with barley HvUGT13248 (UniProt ID: M0Y4P1.1) as the query. This resulted in finding orthogroup OG0000783 which included a total of 89 proteins, 6 of which were from barley and 13 from hexaploid oats. Phylogenetic analysis of this group revealed that barley HvUGT13248, together with six oat proteins and a further barley protein, forms a separate clade (Figure 1). The branch of oat proteins closest to HvUGT13248 in this clade consists of products of three genes: AVESA.00010b.r2.6AG1068650.1, AVESA.00010b.r2.6AG1068570.1 and AVESA.00010b.r2.4CG1255890.1. The first two proteins are close homologues. Their protein sequence comprises 477 amino acids exhibiting 95% sequence identity. The genes are located on chromosome 6A, where they are embedded in a cluster with nine other UGTs (including three from the HvUGT13248-orthogroup). Similar genomic organisation into clusters was found for DON-detoxifying UGTs of several other species, such as *Arabidopsis thaliana*, *Brachypodium distachyon* and rice [50,51,52]. Finally, gene AVESA.00010b.r2.4CG1255890.1 is located on chromosome 4C, and the encoded protein is heavily truncated compared to the other two (Appendix A).

A second branch related to the HvUGT13248 branch is formed by three more oat sequences: AVESA.00010b.r2.5DG0945080.1, AVESA.00010b.r2.4CG1260040.1 and AVESA.00010b.r2.6AG1064420.1, which are located on chromosomes 5D, 4C and 6A, respectively. Notably, these three genes are not clustered with other UGTs.

We have limited our functional characterisation to the two oat genes AVESA.00010b.r2.6AG1068650.1 and AVESA.00010b.r2.6AG1068570.1, considering a combination of the factors, such as their strongest phylogenetic relatedness with the barley gene, their highly shared sequence identity and their genomic location in a cluster. From here on, for the convenience of the reader, we designate genes AVESA.00010b.r2.6AG1068650.1 as ***AsUGT1*** and AVESA.00010b.r2.6AG1068570.1 as ***AsUGT2***.

### 2.2. Oat AsUGT1 and AsUGT2 Genes Are Highly Induced by DON and F. graminearum

To elucidate the potential role of the two proposed candidate oat UGTs in DON detoxification, we studied their transcriptional response to treatment with DON and to the infection with a DON-producing strain of *F. graminearum*, which was isolated from Swedish oat. Both UGT transcripts accumulated strongly following treatment with DON (Figure 2). After 12 h, the transcript levels had increased 2000-fold for the *AsUGT1* gene and 200,000-fold for the *AsUGT2* gene compared to mock treatment. Expression levels of both genes followed a similar pattern, showing an increase in the transcript levels 4 h after application, peaking at 12 h and then declining at 48 h.

Infection of oat spikelets with the 3-ADON chemotype strain of *F. graminearum* resulted in a high expression of both UGTs. The transcripts started to accumulate at 72 h post-inoculation and reached a 1400-fold increase for *AsUGT1* and a 10,000-fold increase for *AsUGT2* at 168 h post-inoculation (Figure 2).

### 2.3. Expression of AsUGT1 and AsUGT2 Confers Resistance to Trichothecenes in Yeast

To validate the DON-detoxifying abilities of the two oat UGTs, the open reading frames were codon-optimised and custom-synthesised. The genes were cloned into a yeast expression vector (with *LEU2* marker expression driven by the strong constitutive *ADH1* promoter), and then transformed into the toxin-sensitive yeast strain YZGA515, as reported previously [53]. Yeast transformants were spotted on plates with yeast peptone dextrose (YPD) media containing five different trichothecenes—DON, NIV, T-2 toxin, HT-2 toxin and DAS (diacetoxyscirpenol). Vectors containing *HvUGT13248* and the *S. cerevisiae* trichothecene-3-O-acetyltransferase (*ScAYT*1) genes were used for transformation as positive controls and the empty vector as a negative control. Figure 3 shows that expression with either *AsUGT1* or *AsUGT2* could confer increased levels of resistance to some of the trichothecenes in yeast. The transformants tolerated DON, NIV and HT-2 (highest concentrations tested were 120 mg/L DON, 80 mg/L NIV and 15 mg/L of HT-2 toxins). In contrast, the strains with the barley and also oat UGTs remained as sensitive as the controls containing the empty vector following treatment with T-2 toxin and DAS. The *ScAYT1*=positive control also inactivated toxins with an acetylated C4-OH. A low, only partially inhibiting concentration of T-2 (0.75 mg/L), which still allowed for growth of the empty vector strain, is shown in Figure 3: the *AYT1* overexpressing strain already shows better growth than the empty vector control. At higher concentrations (not shown), as in the case of DAS, all UGT-expressing strains are fully inhibited, and only the *ScAYT1* transformants grew.

### 2.4. AsUGT1 and AsUGT2 Gene Products Convert DON into DON-3-O-β-D-Glucoside

To investigate the ability of the oat UGT enzymes to form a glucose-conjugate of DON, we aimed at expressing each gene in *E. coli*, purifying the proteins and performing the enzymatic assays. The enzymes were expressed in *E. coli* strain BL21(DE3) as fusion proteins with an N-terminal His6-tag and a maltose-binding protein (*malE*) solubility tag.

Based on SDS gels (total protein extracts after inductions were loaded), both AsUGT1 and AsUGT2 proteins were highly expressed (Figure 4). Unfortunately, neither yielded an active protein preparation suitable for kinetic characterisation after two-step purification (Ni-chelate affinity purification and Sephadex size exclusion chromatography). To test whether the proteins expressed in *E. coli* are active nevertheless, we performed a DON-detoxification assay, with whole cells producing high amounts of the oat UGTs. DON was added directly to bacterial cultures (final concentration 100 mg/L) followed by 16 h incubation with shaking. We relied on the constitutive UDP-glucose of live *E. coli* as a source of co-substrate in the glycosylation reaction, i.e., no external UDP-glucose was added. Cell suspensions with added DON were sampled at two time points (0 h and 16 h), and the resulting extracts were analysed using LC-MS/MS. On average, 25% of DON was converted to D3G in cells expressing AsUGT1 and 22% in cells expressing AsUGT2 (Figure 4). *E. coli* transformed with the empty vector, expressing only His6-tag and MalE, were used as controls. No D3G was detected in these cultures.

## 3. Discussion

Land plants evolved the ability to conjugate endogenous metabolites and also xenobiotics by forming sugar conjugates. The co-substrates are UDP-activated sugars; in plants UDP-glucose is used most frequently, resulting in the formation of beta-glucosides due to the activity of the inverting enzymes of the family 1 of UDP-glycosyltransferases. This gene family of glycosyltransferases acting on small molecules expanded from 1 gene in the unicellular alga *Chlamydomonas reinhardtii* to about 150 genes in diploid plants [54]. For instance, 159 presumably functional genes were identified in *B. distachyon* [52], and 147 genes were identified in maize [55]. The number of 179 UGTs reported in wheat [56] is thus surprisingly low for a hexaploid.

The ability of plants to conjugate trichothecene toxins with a glucose molecule efficiently is seemingly one of the important contributing factors counteracting the fungal virulence factor, which presumably blocks or at least delays expression of defence transcripts produced in response to the pathogen. In the genus *Fusarium*, the ability to produce trichothecenes seems to have existed for at least 27 million years [57], predating the split of barley and the diploid ancestors of hexaploid wheat and oat. One can therefore expect that orthologous genes have taken over the task to counteract the fungal virulence factor. Yet, it is nontrivial to recognize true orthologues in the rapidly evolving gene family, as it seems that UGTs can rapidly increase in copy number, leading to the formation of clusters and the subsequent relocation of individual copies to other chromosomal locations, leading to functional redundancy. Duplicated genes can undergo sequence changes to obtain new specificities toward structural variants of the toxin evolved by the fungus, or to undergo gene death—leading to the multiple truncated genes present in plant genomes.

Previously, it was virtually impossible to find the relevant UGT candidates among hundreds of highly homologous genes coming from three ancestral genomes in oat. Recently, two high-resolution whole oat genome sequences became available for researchers and breeders: OT3098 v2 [47] and cv. Sang [48]. This made it possible for us to identify oat candidate genes involved in the conjugation of trichothecenes.

In this study, we have identified and functionally characterised two homologous oat glucosyltransferase candidates, AsUGT1 and AsUGT2, and verified their trichothecene-detoxifying properties. The candidates were found on the basis of a gene orthology search, where the well-studied barley HvUGT13248 protein was used as a query. Transgenic experiments with the barley *HvUGT13248* gene in wheat and with susceptible barley lines showed that this gene conferred resistance to both DON and NIV, and subsequently, to *F. graminearum* infection in planta [44,45,58]. Interestingly, despite the expected redundancy and overlapping specificity with other UGTs, the loss of function of this gene led to increased toxin susceptibility [46]. Similarly, it has been shown that an overexpressed *B. distachyon* UGT also conferred increased DON and Fusarium resistance, and that the loss of function in a tilling mutant led to both lower toxin and *Fusarium* resistance [59].

We focused on two oat UGTs which are closely related (95% identity) and exhibit high sequence identity to the HvUGT13248 gene product: 78% for AsUGT1 and 76% for AsUGT2. These two genes are located near each other and are part of a larger cluster containing multiple UGT genes. Similar UGT gene clusters were reported for rice and *B. distachyon*, but highly similar genes that were inducible by DON nevertheless showed a lack of substrate specificity towards DON [51,52]. We have also found an UGT homologue (AVESA.00010b.r2.4CG1255890.1) which is located in the same cluster but is unlikely to encode a functional enzyme due to a large truncation (197 aa compared to the length of 477 aa in AsUGT1 and AsUGT2).

In the initial study reporting validation of the function of the barley HvUGT13248 [50], a subset of UGTs inducible by DON and toxin-producing *F. graminearum* were identified, but interestingly only one out of five DON-inducible barley UGT cDNAs conferred DON resistance in yeast. In our study we found that both *AsUGT1* and *AsUGT2* have very low basal expression. Therefore, calculating inducibility factors may be rather misleading since they are based on divisions by small numbers (close to 0). In the induced state, AsUGT2 is expressed at a rate about 10-fold higher than AsUGT1, which is not immediately obvious in Figure 2 due to the log-scale used: in the *Fusarium*-infected samples, the ratio *AsUGT2*/*AsUGT1* after 72 h was 7.6-fold higher, and 10.5× and 6.9× higher after 96 and 168 h, respectively. The inducibility by DON was also higher. The ratio of *AsUGT2*/*AsUGT1* was already 9.5× higher 4 h after treatment and continued to be 14.6× and 16.8× higher at 8 h and 12 h, respectively. Assuming that both enzymes have similar enzymatic properties, as suggested by the results for the DON-detoxification assay shown in Figure 4, this could mean that the *AsUGT2* gene is more important in the interaction with the toxin-producing fungus.

The dynamics of DON-induced expression of the two oat UGTs is similar to the expression of the orthologous genes *Bradi5g03300* in *B. distachyon* [52] and *HvUGT13248* in barley [60]. The molecular mechanism of the DON-induced response seems to be operating at a comparable rate in these closely related monocots. As numerous UGTs compete for the same co-substrate (UDP-glucose), inducibility of the genes with the right substrate specificity against a toxin seems to be of evolutionary importance.

The expression pattern of two oat UGTs in response to *F. graminearum* infection is in a good agreement with the study of *F. graminearum* infection in barley [61], where it was shown that DON production by the fungus increased substantially at 72 h post-infection and kept increasing after 144 h. In our experiments, oat spikelets were infected with an excess of *Fusarium* spores in an environment favourable for the fungus, and the plant’s defence system could not counteract such a strong infection. In such circumstances, *F. graminearum* would continually produce DON at 72 h post-infection so that genes highly inducible by DON were also forced to remain expressed.

Unfortunately, the recombinant enzymes expressed in *E. coli* were recalcitrant to purification, losing activity in vitro very rapidly. Potentially, changing the location and/or type of the purification tag could lead to stable enzymes, allowing for the determination of kinetic properties. Yet, the results from the feeding experiment of intact *E. coli* cells clearly show that the enzymes are active with DON. *E. coli* cells containing similar amounts of proteins (Figure 4a) converted the initial 100 mg/L DON into comparable amounts of D3G (Figure 4b). This DON concentration is not saturating for the HvUGT13248 enzyme (Km 1.5 mM) [62], and most likely reflects that the two oat UGTs have similar specific activity. So, the relative importance should depend on the actual protein levels. As the basal level is very low, and trichothecenes block translation, rapid inducibility could be decisive for the outcome: whether the induced and translated UGT transcripts can detoxify DON, or DON is able to block the expression of the detoxification enzyme. In the absence of knowledge on the protein levels, it remains unclear whether a higher basal expression level (leading to a lower induction coefficient) is of advantage or not. Proteomics experiments should be informative to address this question.

Oat is much more prone to infection by *Fusarium* species other than *F. graminearum* compared to barley and wheat, and to the accumulation of other trichothecenes more prevalent in oat. We therefore wished to determine whether the two oat UGTs have a similar or different substrate specificity for different relevant trichothecenes. Oat UGTs, which resemble HvUGT13248, conferred resistance to DON, NIV and HT-2. Seemingly, as also demonstrated for HvUGT13248, acetylation of the C4-OH in T-2, DAS and Fusarenone X (data not shown) allows the fungal toxins to escape detoxification by the UGTs.

Site-directed mutagenesis of UGT Os79 from rice showed that increasing the volume of the trichothecene-binding pocket could improve the activity of the enzyme for the Type A trichothecenes which possess an acetyl-group at the C4 atom [63]. Yet, plants overcome this problem by efficiently deacetylating these toxins by carboxylesterases [64], which is seemingly also the case in oat [43]. The Norwegian oat cultivar Odal, which was initially selected for its low DON-accumulating properties, was later found to accumulate as high levels of T-2/HT-2 during infection with *F. langsethiae* as other susceptible cultivars [65]. This suggests that there could be different mechanisms behind DON and T-2/HT-2 resistances, and although obviously AsUGT1 and AsUGT2 at least have the ability to glycosylate HT-2, variation in the T-2 deacetylation capability might exist and limit detoxification by glycosylation.

The roles of different trichothecenes as virulence factors and resistance mechanisms in oat are largely unknown. Recent screening trials resulted in the finding of a number of FHB-tolerant and FHB-susceptible oat cultivars [66,67,68,69]. In these trials, DON content in oat kernels was measured as one of the traits of disease resistance. Complementing these studies with the analysis of the D3G/DON ratio and the ratios of other toxins and their glucosides could give an indication of differences in the glycosylation abilities of the cultivars. Analysing the polymorphism of the AsUGT1 or AsUGT2 across the cultivars with enhanced or weakened DON conjugation would help to define the role of these genes in the resistance of oat against FHB.

Another aspect of the structural differentiation of plant UGTs is their interaction with potential fungal inhibitors. Culmorin often co-occurs with DON in *Fusarium*-damaged cereals [70] and has been shown to have a synergistic phytotoxic effect together with DON [71] (Michlmayr et al., in preparation). Furthermore, it was found to act as a powerful inhibitor for some plant UGTs but not for others (Michlmayr et al., in preparation). Testing the ability of oat UGTs orthologous to HvUGT12348 to withstand inhibition by culmorin could help to understand the role of diversity and the apparent redundancy of UGTs in trichothecene detoxification.

## 4. Conclusions

In this study, we have characterised the first oat UGT genes encoding enzymes capable of inactivating different trichothecenes as an approach to improving our understanding of the molecular mechanisms of *Fusarium* resistance in oat. Further experiments, both with recombinant UGT proteins and transgenic/edited plants, should reveal their (redundant) role in *Fusarium* infections and in the accumulation of mycotoxins and their masked forms. In future work, the two reported UGT genes could be used as markers for screening and breeding FHB-resistant oat germplasm, which should eventually result in low mycotoxin levels in the end product.

## 5. Materials and Methods

### 5.1. In Silico Analysis

To identify oat candidate UGTs, we searched the oat and barley proteins present in the orthogroups reported by Kamal et al. [48]. Oat sequences, gene annotations and orthogroups are available at https://doi.ipk-gatersleben.de/DOI/43ec5a99-d7b6-4a28-b9d8-4a6ec81a60fd/bc36ea85-b944-4bc3-a52f-f78596335ea5/2, (accessed on 29 May 2022), and the barley sequences were downloaded from https://doi.ipk-gatersleben.de/DOI/b2f47dfb-47ff-4114-89ae-bad8dcc515a1/7eb2707b-d447-425c-be7a-fe3f1fae67cb/2, (accessed on 29 May 2022) separately using diamond v2.0.0 [47] parameters—max-target-seqs 1) with barley HvUGT13248 (UniProt ID: M0Y4P1.1) as the query. The oat and barley sequences in the identified orthogroup were aligned using MUSCLE (v3.8.1551) [72], and phylogenetic trees were constructed using fasttree (v2.1.10) [73], with tree visualisation performed using iTOL [74]. FASTA files of the oat and barley sequence alignments of the orthogroup OG0000783 proteins (Appendix A) and the alignment of 4 full barley and 13 full oat FASTA files of OG0000783 proteins (Appendix A) are available in the Appendix A.

### 5.2. Plant and Fungal Material

Four seeds per pot (16 cm in diameter) of oat cv. Belinda were planted in a mix containing soil (Krukväxtjord med lera och kisel, SW Horto, Sweden), 9% (*v*/*v*) perlite and 0.3% (*v*/*v*) Basacote plus 3M granulated fertilizer (N-P_2_O_5_-K_2_O(+MgO+S) 16-8-12(+2+5) (Compo Expert, Münster, Germany). Plants were grown in a growth chamber for 16 h during the day at 22 °C and 6 h at night at 18 °C. Light intensity was set at 300 µE m^−2^ s^−1^ at the panicle level, and relative humidity was kept at 65%.

The *F. graminearum* strain LS_G2 (3-ADON chemotype based on genotyping according to Quarta et al., 2006 [75]) was isolated from oat growing in Sweden. Inoculum was obtained by culturing on potato dextrose agar (PDA) plates for 5 days and subsequently grown in a liquid carboxymethylcellulose (CMC) medium (1.5% CMC, 0.1% NH_4_NO_3_, 0.1% KH_2_PO_4_, 0.05% MgSO_4_ 7H_2_O, and 0.1% yeast extract) containing 100 units of penicillin and 0.1 mg streptomycin (Sigma-Aldrich, Lyon, France) per 1 mL of medium for another 5 days at 25 °C and 150 rpm shaking. Macroconidia were filtered through 100 µm cell strainer (Sarstedt, Nümbrecht, Germany), harvested by centrifugation at 4000 rpm for 10 min and washed once with sterile distilled water. Spores were resuspended in sterile water with added Tween20 (Sigma-Aldrich, Lyon, France) at a final concentration of 0.02%. Concentration of macroconidia was adjusted to 100,000 spores/mL.

### 5.3. Plant Treatment with DON and F. graminearum

A DON solution (0.2 mg/mL) was prepared by dissolving DON (Sigma-Aldrich, Lyon, France) in deionised water. For the analysis of the DON-induced expression of UGT genes, plants were treated at anthesis (Zadoks 65) either with aqueous DON solution or with water (mock). At time point zero, 10 µL of DON solution was pipetted in the space between two adjacent florets in a spikelet. Inoculated panicles were covered with 3 L plastic bags, sprayed with distilled water beforehand. A total of 10 spikeletes were treated per panicle. At 0, 4, 8, 12, 24 and 48 h post-treatment, three replicates per time point were sampled and immediately frozen in liquid nitrogen.

Similarly, for the analysis of the *F. graminearum*-induced expression of UGT genes, plants were treated at anthesis (Zadoks 65) either with a fungal spore suspension or with water containing 0.02% Tween20 (mock). At time point 0, 20 µL of spore suspension (100,000 spores/mL) was pipetted in the space between two adjacent florets in a spikelet. Inoculated panicles were covered with 3 L plastic bags, misted with distilled water beforehand. After 72 h the bags were removed, and panicles were sprayed with distilled water three times per day. In total, 10 spikelets were treated per each panicle. A total of 3 replicates of samples were collected at 0, 24, 48, 72, 96, 120, 168 h post-treatment, and frozen immediately in liquid nitrogen.

### 5.4. Analysis of DON and F. graminearum-Induced UGT Expression in Oat

Three experimental repetitions were performed. In each, RNA was extracted from two spikelets. Separate spikelets were grounded under frozen conditionand pre-cooled in liquid nitrogen plastic tubes (Sarstedt, Nümbrecht, Germany) and two 5 mm stainless steel beads (Qiagen, Hilden, Germany) using Precellys Evolution homogenizer (Bertin Technologies, Montigny-le-Bretonneux, France) as follows: 25 s homogenisation at 5500 rpm, cooling in liquid nitrogen, repeated homogenisation. RNA was extracted from 2 pooled homogenised spikeletes using RNeasy Plant Mini kit (Qiagen, Hilden, Germany). The quality and quantity of the extracted RNA were analysed spectrophotometrically and by agarose electrophoresis. DNA was removed from the samples by using the DNA-free Kit (Invitrogen, Waltham, MA, USA) according to the manufacturer’s instructions. cDNA synthesis was performed using the iScript cDNA synthesis Kit (Bio-Rad Laboratories, Hercules, CA, USA). cDNA samples were diluted 1:20 with TE buffer (Tris-HCl, pH 8.0 10 mM, EDTA 1 mM), aliquoted and stored at −20 °C. Primers and probes 5′ 6-FAM/ZEN/3′IMFG (Integrated DNA Technologies, Leuven, Belgium) were designed to amplify fragments of the gene transcripts of AVESA.00010b.r2.6AG1068570.1 and AVESA.00010b.r2.6AG1068650.1 (for primer sequences see Appendix A) For the reference target of the oat tubulin-alfa gene, a pair of primers and a probe 5′HEX/ZEN/3′IBFQ (Integrated DNA Technologies, Leuven, Belgium) were designed. Duplex qPCR reactions targeting one of the UGTs and a reference gene were performed in 15 µL final volume, containing 2 µL of the diluted cDNA product, 750 pmol of each primer, 225 pmol of each probe and 7.5 µL PrimeTime™ Gene Expression Master Mix (Integrated DNA Technologies, Leuven, Belgium). Reactions were carried out in a technical duplicate using a CFX96 Real-Time qPCR system (Bio-Rad Laboratories, Hercules, CA, USA). The Ct was automatically determined for each reaction by a Bio-Rad system set with default parameters. The comparative ΔΔCt method was used to evaluate the relative quantities of each amplified product in the samples [76].

### 5.5. Cloning and Yeast Expression of Two Oat Full-Length UGT cDNAs

Full-length cDNA sequences corresponding to AVESA.00010b.r2.6AG1068570.1 and AVESA.00010b.r2.6AG1068650.1 were codon-optimised for expression in *S. cerevisiae* with the addition of flanking HindIII and NotI restriction enzyme sites (Appendix A. Modified sequences were custom synthesised as gBlocks fragments (Integrated DNA Technologies, Leuven, Belgium). Expression plasmids were constructed by digesting the custom synthesised genes with HindIII and NotI and ligating the fragments into the backbone of pWS1921 [50] cleaved with the same enzymes. The resulting plasmids contain the synthetic genes c-terminally fused to the c-myc epitope under the control of the strong constitutive ScADH1 promoter and terminator. UGT gene sequences were verified by Sanger sequencing (Eurofins Genomics, Ebersberg, Germany) using specific primers for the vector backbone. UGT expression vectors, the empty vector pBP910 as a negative control and plasmids expressing *HvUGT13248* [50] and *ScAYT1* as positive controls were transformed into the toxin-sensitive yeast strain YZGA515. Yeast *ScAYT1* encodes a 3-OH trichothecene acetyl transferase that converts DON into a less toxic 3-ADON [77]. Transformants were selected on a synthetic complete medium lacking leucine. Exponentially growing cultures were diluted to an optical density (OD600) of 0.05 and 0.01 with the fresh selective medium. A total of three µL of these dilutions in two replicates were spotted onto YPD plates containing different concentrations of DON, NIV, T-2, HT-2 toxin and DAS. Plates were incubated at 30 °C for 3 days.

### 5.6. Expression of Recombinant Oat UGTs in E. coli

Two oat UGT cDNAs were cloned into plasmid pJW1 using HindIII and NotI restriction sites. pJW1 was derived from pCA02 [62] by digestion with EcoRI, Klenow fill-in and relegation, so that the yeast vector inserts can be shuffled into the vector in frame with the N-terminal His6-tag followed by a maltose binding protein (His6-MalE-UGT) in *E. coli* strain BL21(DE3). Protein expression was carried out in terrific broth (TB) medium supplemented with 100 mg/L ampicillin. Isopropyl-β-D-1-thiogalactopyranoside (IPTG) (Sigma-Aldrich, Lyon, France) was added at a final concentration of 1 mM when the optical density (OD600) of the bacterial culture reached 0.5–0.8. The cultures were further incubated for 19 h at 20 °C and 100 rpm. Bacterial biomass was harvested by centrifugation at 4000× *g* for 20 min and resuspended in 50 mM potassium phosphate buffer pH7 with 10% (*w*/*v*) glycerol and 1% (*w*/*v*) Triton X-100. The cells were disrupted with Q500 sonicator (Qsonica, Newtown, CT, USA). The cell extract was clarified by centrifugation at 14,000× *g* for 30 min. Protein purification was performed by immobilised metal affinity chromatography on Ni+2 charged 5 mL Hi-Trap Chelating HP column (GE Healthcare, Uppsala, Sweden). The prior loading on the column, NaCl and imidazole were added to the cell extract, at final concentrations of 500 mM and 25 mM, respectively. The loaded column was washed with a buffer containing 50 mM potassium phosphate pH 7.0, 500 mM NaCl and 25 mM imidazole. Bound protein was eluted with the same buffer containing 500 mM imidazole. The fractions, containing the protein, were pooled, and the buffer was changed to 50 mM potassium phosphate pH7.0, 50 mM KCl and 10% (*w*/*v*) glycerol by using gel filtration columns PD-10 with Sephadex G-25 resin (GE Healthcare, Uppsala, Sweden). Concentration of the protein was determined using Bradford reagent (Sigma Aldrich, Lyon, France). Samples obtained during the protein purification process were analysed with Sodium dodecyl sulfate polyacrylamide gel electrophoresis (SDS-PAGE).

### 5.7. Glucosyltransferase Activity Assay

Reactions were performed at 25 °C in 100 mM Tris-Cl pH 7, and 2 different concentrations of DON in the reactions were tested. Low DON concertation reactions contained 30 µM DON (Sigma Aldrich, Lyon, France), 1 mM UDP-glucose (Uridine-diphosphate-glucose disodium salt) (Sigma Aldrich, Lyon, France) and 1 mg/mL purified enzyme. High DON concertation reactions contained 1 mM DON, 10 mM UDP-glucose and 1 mg/mL purified enzyme. Preparations of proteins expressed by *E. coli* transformed with the empty vector were used as controls. All reactions were performed in three repetitions. Samples from the reactions were taken at 0, 1, 2, 4, 6, 24 h after the start, and the reactions were stopped by adding 10 volumes of 99.8% methanol (Sigma-Aldrich, Lyon, France). Products of the reactions were analysed with quantitative LC-MS/MS.

### 5.8. Whole Cell Treatment with DON

Three transformants of *E. coli* strain BL21(DE3) carrying pJW1 with either *AsUGT1* or *AsUGT2* and two transformants carrying empty vectors expressing only His6-tag and MalE were grown in 50 mL of terrific broth (TB) medium with an added 100 mg/L ampicillin. Isopropyl-β-D-1-thiogalactopyranoside (IPTG) (Sigma-Aldrich, Lyon, France) was added to a final concentration of 1 mM when the optical density (OD600) of the bacterial culture reached 0.5–0.8. The cultures were further incubated for 18 h at 20 °C and 100 rpm.

1.8 mL from each bacterial culture was harvested by centrifugation for 3 min at 10,000 rpm and at RT; the pellet was washed once with M9 media with an added 100 mg/L ampicillin. The cells were resuspended in 1.5 mL of M9 media with an added 100 mg/L ampicillin. 900 µL of each cell suspension was transferred into 14 mL round-bottom tubes (Corning 352057, Corning Life Sciences, Corning, NY, USA), and aqueous DON solution (Sigma Aldrich, Lyon, France) was added to the suspensions at a final concentration of 100 mg/L. The controls consisted of bacterial cultures with added distilled water instead of DON. Samples were collected at 0 h and 16 h after incubation at 22 °C with 100 rpm shaking. A total of 2 volumes of 99.8% methanol (Sigma-Aldrich, Lyon, France) were added to the collected bacterial suspensions. Samples were kept at 4 °C until further analysis. Prior to the LC-MS/MS analysis, samples were centrifuged for 15 min at 13,000 rpm and RT, and supernatants were collected and diluted five times with 99.8% methanol.

Remaining *E. coli* suspensions (not used in the DON treatment assay) were used in the Sodium dodecyl sulfate polyacrylamide gel electrophoresis (SDS-PAGE) analysis. Crude extract samples were prepared by adding sample loading buffer (100 mM Tris-HCl pH 6.8, 4% (*w*/*v*) SDS, 0.2% (*w*/*v*) bromophenol blue, 5% (*v*/*v*) β-mercaptoethanol, 20% (*v*/*v*) glycerol) to the equal volume of *E. coli* suspensions and incubating at 95 °C for 5 min. The samples were loaded on a precast Criterion TGX gel (8–16%) (Bio-Rad), and the electrophoresis was performed using a Criterion cell system (Bio-Rad Laboratories, Hercules, CA, USA). PageRuler^TM^ unstained protein ladder (10–200 kDA range) (Thermo Scientific) was used as a molecular mass marker. Coomassie blue staining was performed to visualise the protein bands.

### 5.9. Quantitative LC-MS/MS

The LC-MS/MS analysis was performed using the service facility of the Centre of Analysis and Synthesis at the Department of Chemistry, Lund University. The analysis was performed on am Agilent Triple Quadrupole ESI-QqQ 6495B system coupled to an ultra-high performance liquid chromatography (UHPLC) Agilent 1290 Infinity II (Agilent Technologies, Waldbronn, Germany). For HPLC separation, Acquity CSH C18, 2.1 × 100 mm i.d., 1.7 µm particle size (Waters, Dublin, Ireland) analytical column was used with flow rate 0.5 mL/min, at 60 °C and with injection volume 1 µL, and the flow went to the electrospray source. The eluent A was water, and the eluent B was 95% methanol, both containing 10 mM ammonium acetate. The separation was performed as follows: 0–6.5 min: 10% B, 6.5–8.0 min: gradient of 10–100% B, column flushing 8.0–9.0 min 100% B, 9.0–9.5 min 100–10% B, column equilibration 9.5–12.5 min 10% B.

The MS parameters were set at as follows: The AJS electrospray ion force was operated in negative mode capillary voltage 3000 V, nozzle voltage 1500 V, gas temperature 200 °C, gas flow 14 L/min, nebulizer 20 psi, sheath gas temperature 250 °C, sheath gas flow 11 L/min. The ion unnel parameters were set to 90 V and 60 V for high pressure and low pressure, respectively. Fragmentor was set to 380 V and cell acceleration to 5 V.

For quantification using external calibration standards at concentrations of 1, 2, 4, 6, 8, 10 and 20 µg/mL, DON (Sigma-Aldrich, Lyon, France) and DON-3-O-glucose (Sigma-Aldrich, Lyon, France) were injected in acetonitrile. DON was detected at 3.0 min after the transition *m*/*z* 355.2 → 59.1 (quantifier, CE 30) and 355.2 → 264.9 (CE 15, qualifier), while DON-3-O-glucose was observed at 3.7 min and observed at *m*/*z* 517.2 → 427.1 (CE30, quantifier) and *m*/*z* 517.2 → 58.9 (CE 60, qualifier). Appendix A contains images of the chromatograms of the substrate and the products of the reactions.

## Figures and Tables

**Figure 1 toxins-14-00446-f001:**
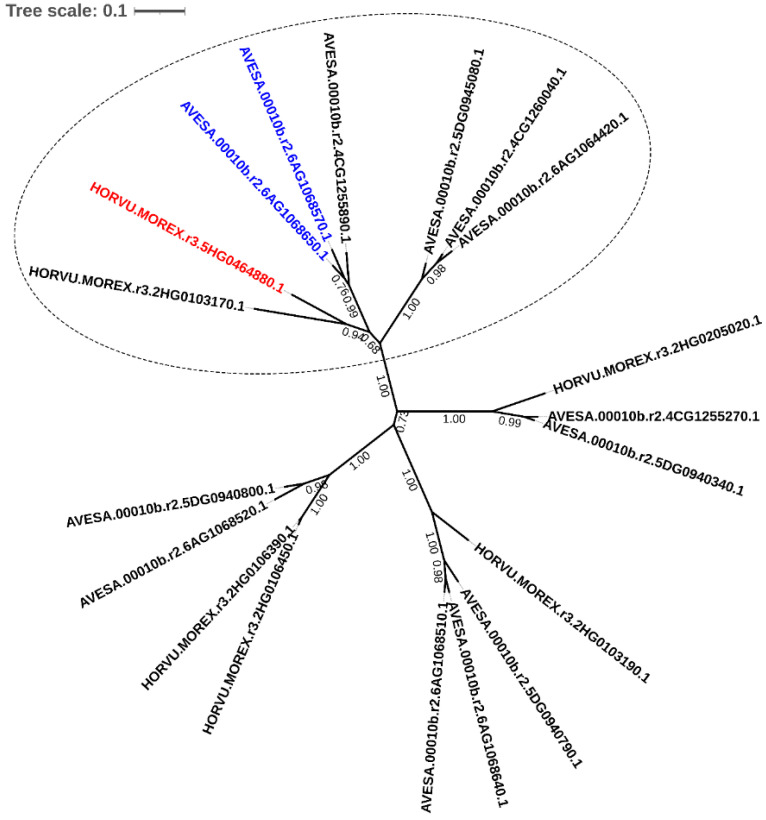
Phylogenetic analysis of putative oat trichothecene-detoxifying UGT proteins together with barley orthologs present in orthogroup OG0000783. HORVU.MOREX.r3.5HG0464880.1 corresponding to barley HvUGT13248 is marked in red. The two oat proteins, AVESA.00010b.r2.6AG1068650.1 (AsUGT1) and AVESA.00010b.r2.6AG1068570.1 (AsUGT2), chosen for the current study are marked in blue. The clade formed by the proteins phylogenetically closest to HvUGT13248 is marked with a dotted line.

**Figure 2 toxins-14-00446-f002:**
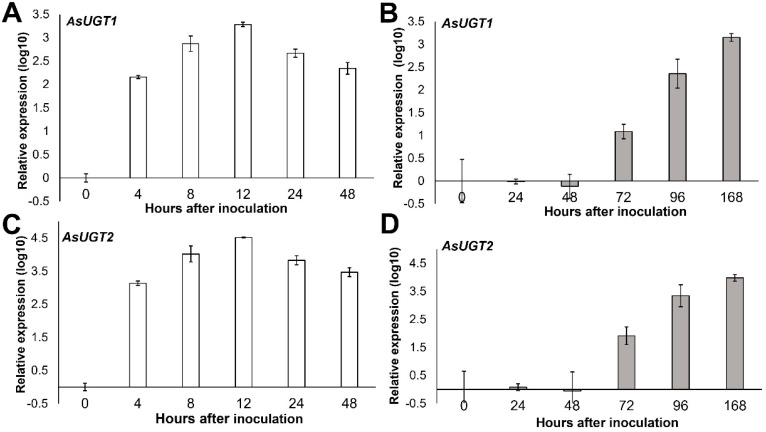
Relative expression of *AsUGT1* and *AsUGT2* genes in oat spikelets after inoculation with either DON (**A**,**C**) or *F. graminearum* (**B**,**D**).

**Figure 3 toxins-14-00446-f003:**
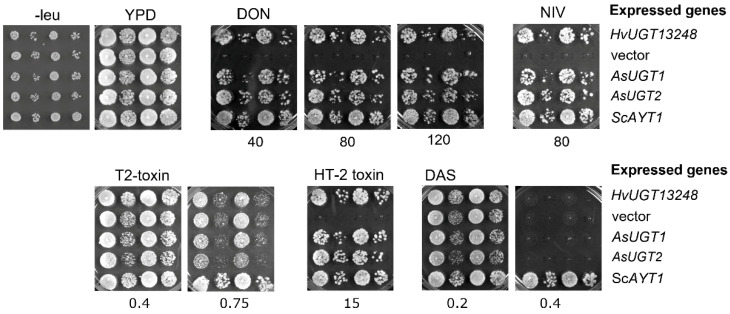
Spottings of yeast transformants expressing oat glucosyltransferases *AsUGT1* and *AsUGT2* on plates with indicated concentrations of five different mycotoxins. Strains carrying barley *HvUGT13248*, yeast acetyltransferase *ScAYT1* and the empty vector were used as controls. Toxin-containing plates are based on YPD. Control plates without toxin are SC-leu, where only transformed yeast cells can grow, and the rich medium YPD, which allows for growth of strains without a plasmid. Two independent transformants of each construct were spotted in two different dilutions.

**Figure 4 toxins-14-00446-f004:**
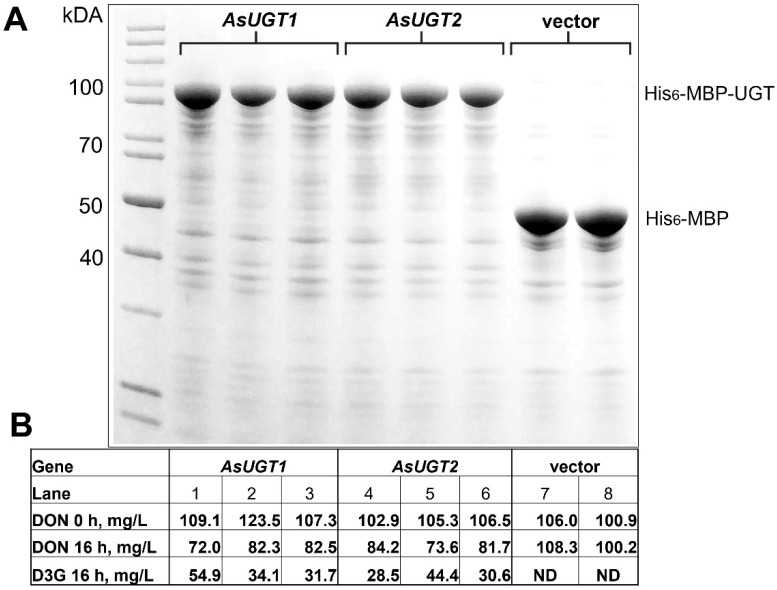
(**A**) SDS-PAGE analysis of crude extracts of independent transformants of *E. coli* BL21(DE3) cells expressing either AsUGT1 (lines 1–3) or AsUGT2 (lines 4–6) as fusion proteins with the N-terminal His-tag and maltose binding protein (MalE). Transformants carrying a empty vector with only His-tag and MalE were used as a control (lines 7–8). (**B**) Concentrations of DON and DON-3G at 0 h and after 16 h incubation with *E. coli* transformants containing the expression vector with the *AsUGT1* gene (1–3), *AsUGT2* gene (4–6) or the empty vector (7–8).

## Data Availability

Oat sequences, gene annotations, and orthogroups are available at https://doi.org/10.5447/ipk/2022/2, accessed on 1 June 2022, barley sequences downloaded from https://doi.org/10.5447/ipk/2021/3, accessed on 1 June 2022.

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
