# Peer review of "Identification and Functional Characterisation of Two Oat UDP-Glucosyltransferases Involved in Deoxynivalenol Detoxification"

_toxins, 2022, doi:10.3390/toxins14070446_

Round 1

Reviewer 1 Report

The manuscript reports a very interesting study on a mechanism of Fusarium resistance in oat. In particular, two candidate genes have been in silico identified and deeply studied using different approaches, such as the expression profiles in plant and their characterization in transformed yeast. The work is presented in a very clear way and it is easy to follow the workflow and the rationale behind.

I have only a minor suggection about Figure 2: please improve the readibility of the figure using bigger characters.

In conclusion, in my opinion, the work is relevant for improving Fusarium control in oats, a key factor for a cereal with increasing interest for human nutrition.

Author Response

Thank you very much for finding time to review our manuscript and for the positive and supportive feedback. Indeed, the font size on Figure 2 was way too small. We have now enlarged all the font size consistently with at least 4 points in hope that it would become more viewer-friendly.

Kind regards,

the authors

Reviewer 2 Report

This work identified and characterized two ota UDP glucosyltransferases. The enzyme is proved that overexpression in wheat could increase resistance to deoxynivalenol and NIV, and decrease disease severity of FHB. In the abstract, the authors mentioned some other fusarium mycotoxins, including T-2, HT-2. However, only DON was investigated in this work. This is confusing.

The authors mentioned DON could be converted to DON-3-O-glucoside after treatment. However, there are no related information for identification of DON-3-O-glucoside. There are also no related information about DON, DON-3-O-glucoside, or other related materials information.

Author Response

Thank you very much for finding time to revise our manuscript. We apologize for causing confusion in the interpreting the results of experiments with mycotoxins.

Previously studied plant UDP-glucosyltransferases from rice, brachypodium and barley  showed to have an affinity for several different trichothecenes. Similar to those studies, we have chosen to evaluate oat UGTs in regard to their ability to detoxify trichothecenes with or without acetylation of the C4-OH. Therefore, we have performed spotting assays with 5 different mycotoxins (Figure 3) with corresponding text lines 165-185. We have tested DON, NIV and T-2 toxin (non-acetylated C4-OH) and  HT-2 and DAS (acetylated C4-OH).  These experiment showed that both oat UGTs were behaving similar to barley HvUGT13248: yeast cells, expressing oat UGTs were able tolerate non-acetylated trichothecene (specially DON and NIV) in much higher concentrations compare to the acetylated ones. We discuss the results of that experiments in lines 319-335.

In order to confirm the affinity of the oat enzymes for the trichothecenes we performed whole cell assay with DON. After 16h of incubation substantial part of DON was converted into D3G, which we were able to detect with LC-MS/MS. This experiment confirmed the fact that D3G is indeed the product of activity of cloned oat UGTs. Thank you for pointing out that more detail is needed describing the experimental part and the results of this experiment.

In the revised version of the manuscript, we provide more experimental details on the E.coli cells assay as well as on LC-MS/MS analysis,  such as chromatograms and retention times of the analytes.

We hope that we have managed to address all the points suggested by you in a satisfactory way.

Kind regards,

the authors

Reviewer 3 Report

In this work, the authors outline the identification of two UGT enzymes from oat that are capable of providing resistance against most trichothecenes, notably DON.

All the data and evidence reported here is concise and well-demonstrated. The relevance is also very relevant as this information can provide important data towards breeding more resistant varieties.

One unfortunate part of this work (as the authors themselves noted) is the inability to maintain enzymatic activity in purified enzymes. This could have provided some clear comparisons between these enzymes and others that have previously been described.

On this note, although the authors described how activity in purified protein was lost rapidly, they did not provide any meaningful experimental details of their in-vitro incubation. Even though it didn't work, it must be provided to at the very least, help other researchers understand where improvements could be made.

1) For this assay, was any UDP-glucose included in the incubations? Without this, the reaction would not be very effective. (please address in detail and add to the methods section)

2) Please provide more experimental details regarding the LC-MS/MS analysis. Specifically, the retention times and transitions of the analytes. Chromatograms of the substrate and products compared to standards are needed to support that this is D3G. Are there any other isobaric peaks with D3G that suggests either of these enzymes has the ability to also glycosylate the 15- or 7- position to a much lesser degree?

Minor comments:

i) Overall, it does not seem that expressing these plant proteins in e. coli worked very well. Could same experiment be repeated using the Yeast? Only 22% conversion is also quite low and I wonder if the yeast is able to express a more functional enzyme

ii) Line 285 – Semi-quantitative results shown in Figure 4. Should this actually be referring to Figure 3?

iii)Line 304 = Loosing should be losing.

iv)  Figure 4 – Label the lanes with the Genes so it is easier for the ready to see that expression and activity are similar.

Author Response

Thank you very much for reviewing our manuscript and for the thoughtful suggestions on improving it. We realise that we have, indeed, overlooked some important details of the E.coli expression experiment and LC-MS/MS analysis. In the revised version we have added more information on both.  In supplementary materials we have added a file with chromatograms of the substrate and the product. In the materials and methods section, we have described enzymatic assays and whole cell treatment with DON in more detail.

Regarding the experiment with live E. coli cells – we have not added UDP-glucose to the assay, as  we have relied on the bacteria’s ability  make their own UDP-glucose. Besides, E. coli is known to have a periplasmatic enzyme, ushA  (https://biocyc.org/gene?orgid=ECOLI&id=USHA-MONOMER), which would hydrolyse externally added UDP-glucose,  so that it would not be available for the cytosolic glucosyltransferase enzyme anyway. (As UDP-glucose is a rather costly substance, adding it to the reaction would only increase the cost of this assay without much effect on the glucosylation)

We agree that UDP-Glucose and other UDP-sugars may be limiting the glucosylation reaction in living E. coli, but the goal of that particular experiment was not to optimize production of D3G by whole cells but only to test substrate specificity.

We don’t quite agree that the expression in E. coli did not work well – the SDS-PAGE (Figure 3) illustrates that there is substantial amount of the enzyme produced by E.coli. These are crude extracts (not purified protein), which we should have mentioned in the initial version of the manuscript. Even if most of the produced enzyme might become inactive/insoluble, there is definitely more enzyme produced in E.coli than could be produced in yeast.

Yeast expression was tested with barley enzyme HvUGT13248 in [Schweiger W, Boddu J, Shin S, Poppenberger B, Berthiller F, Lemmens M, Muehlbauer GJ, Adam G. Validation of a candidate deoxynivalenol-inactivating UDP-glucosyltransferase from barley by heterologous expression in yeast. Mol Plant Microbe Interact. 2010 Jul;23(7):977-86. https://doi.org/10.1094/MPMI-23-7-0977]. In this work the D3G conjugate accumulated in the supernatant of DON-treated yeast transformants, but the efficiency was much lower than in E. coli (and this is not due to the ability of the yeast beta-glucosidase to hydrolyse D3G).  In that experiment 500 ppm DON was added to the yeast cells and 700 ppb D3G was produced, i.e. the conversion rate was extremely low. Compared to these values,  22% (33 to 18% in the individual transformants) conversion efficiency of DON into D3G in E.coli seems to be quite high.

As to glucosylation at 15 and 7 position of trichothecenes , when it comes to oat, in the study by [Meng-Reiterer, J.; Bueschl, C.; Rechthaler, J.; Berthiller, F.; Lemmens, M.; Schuhmacher, R. Metabolism of HT-2 toxin and T-2 toxin in oats. Toxins 2016, 8, 1–22. https://doi.org/10.3390/toxins8120364] glucosylation was shown to happen only at the position 3 (with main product of glucosylation being HT2-3-O-β-D-Glc, along with some acetylated at position 15 forms).

Similarly, in DON inoculated wheat no 15 and 7 glucosylation was found:

[Kluger, B.; Bueschl, C.; Lemmens, M.; Michlmayr, H.; Malachova, A.; Koutnik, A.; Maloku, I.; Berthiller, F.; Adam, G.; Krska, R.; et al. Biotransformation of the mycotoxin deoxynivalenol in fusarium resistant and susceptible near isogenic wheat lines. PLoS One 2015, 10. https://doi.org/10.1371/journal.pone.0119656]

In vivo and in vitro analysis of DOGT1 from A.thaliana proved that it “catalyses

the transfer of glucose from UDP-glucose specifically to the 3-OH position of DON” [Poppenberger, B.; Berthiller, F.; Lucyshyn, D.; Sieberer, T.; Schuhmacher, R.; Krska, R.; Kuchler, K.; Glössl, J.; Luschnig, C.; Adam, G. Detoxification of the Fusarium Mycotoxin Deoxynivalenol by a UDP-glucosyltransferase from Arabidopsis thaliana. J. Biol. Chem. 2003, 278, 47905–47914. https://doi.org/10.1074/jbc.M307552200].

Above mentioned and several other works assured us that,  similar to other plant UGTs, oat UGTs would be conjugating DON at the position 3 and didn’t look for other position conjugates.

We have corrected spelling errors and typos. Figure 4 has been improved by  making and  extra line in the table and labelled the lanes (Thank you for pointing all that out).

As to:

ii) Line 285 – Semi-quantitative results shown in Figure 4. Should this actually be referring to Figure 3?

(line 287 in the current version). We are referring to the values in Table in Figure 4 – two enzymes seem to have quite similar activity, judging by the amount of DON converted into D3G. 

We hope that we have managed to address all the points suggested by you and provided satisfactory answers to your questions.

Kind regards,

the authors.

Round 2

Reviewer 2 Report

Detoxification is an important research project for mycotoxins. This work aims to investigate the enzyme for DON detoxification in OAT.  

Based on the corrects of this work, I think it could be accepted in the current version. To make the work better, the importance and potential significance for mycotoxin control could be emphasized.